# Label-free spatio-temporal monitoring of cytosolic mass, osmolarity, and volume in living cells

Daniel Midtvedt[1], Erik Olsén[1], Fredrik Höök[1] & Gavin D.M. Jeffries [2]

Microorganisms adapt their biophysical properties in response to changes in their local environment. However, quantifying these changes at the single-cell level has only recently become possible, largely relying on fluorescent labeling strategies. In this work, we utilize yeast (*Saccharomyces cerevisiae*) to demonstrate label-free quantification of changes in both intracellular osmolarity and macromolecular concentration in response to changes in the local environment. By combining a digital holographic microscope with a millifluidic chip, the temporal response of cellular water flux was successfully isolated from the rate of production of higher molecular weight compounds, in addition to identifying the produced compounds in terms of the product of their refractive index increment $\left(\frac{dn}{dc}\right)$ and molar mass. The ability to identify, quantify and temporally resolve multiple biophysical processes in living cells at the single cell level offers a crucial complement to label-based strategies, suggesting broad applicability in studies of a wide-range of cellular processes.

[1] Department of Physics, Chalmers University of Technology, Gothenburg, Sweden. [2] Department of Chemistry, Chalmers University of Technology, Gothenburg, Sweden. Correspondence and requests for materials should be addressed to D.M. (email: midtvedt@chalmers.se)

The interior of biological cells is a continuously changing environment. Their cytosolic composition modulate in relation to the cell cycle[1], disease state[2–5], as well as in response to changes in the external environment[6,7]. Such modulations include, among others, changes in metabolic activity, formation of phase-separated cytosolic domains[8], increased cytoplasmic stiffness[6], and uptake and release of biomolecular compounds[9]. Physiological responses to changes in the external environment serve partly to protect cells from potentially deteriorating changes in the extracellular environment, however it is also believed that the physicochemical structure of many microbial species is altered in response to changes in the external milieu to favor colonization[10,11].

Traditional ensemble-averaged biomolecular techniques, such as immune assays, genetic screening, and mass spectrometry, have been incredibly successful in identifying the fundamental pathways involved in the cellular response to changes in the external environment. Although powerful, such methods tend to suffer from poor temporal resolution, and do not provide information about population heterogeneity, changes in cellular morphology and other biophysical changes, all of which are likely crucial for the survival and proliferation of individual cells, as well as the colony as a whole. Despite the importance of quantifying and understanding the underlying processes, measuring the corresponding biophysical parameters (e.g. volume, cell mass, and mechanical properties) at the single-cell level has only recently become possible owing to advances in microfabrication and imaging[12]. Nonetheless, measuring these cell parameters with molecular specificity under physiological conditions, in a time-resolved manner, remains a challenge.

Quantitative phase imaging (QPI), measuring the phase shift of light passing through a specimen, has recently emerged as a promising method to study intracellular composition[13]. For a biological cell, the phase shift $\phi$ relates the wavenumber $k$ of the incident light to the cell thickness $z$ as

$$\phi = kz\Delta n, \tag{1}$$

with $\Delta n = n_{cell} - n_{med}$ being the difference in cell refractive index (RI) and medium RI (in this expression, the effects of light refraction at the cell–medium interface are neglected, see Supplementary figure 3). Considering that the RI of a biomolecular solution is linearly related to the mass concentration $c$ (g ml$^{-1}$) of its components, the phase shift integrated over the area of the cell is proportional to cell mass $m_{cell}$[12]. Specifically, one has

$$\Phi \equiv \int \phi(x,y)\mathrm{d}x\mathrm{d}y = kV_{cell}\Delta n = km_{cell}\left(\frac{\mathrm{d}n}{\mathrm{d}c}\right), \tag{2}$$

where $V_{cell}$ is the cell volume, $\left(\frac{\mathrm{d}n}{\mathrm{d}c}\right)$ is the differential change in RI due to a change in biomolecular concentration (typically called the specific RI increment). Here the symbol $\Phi$ is introduced as a shorthand for the integral of the phase shift over the area $A$ that the cell occupies in the microscopy image. The RI increment of biomolecules is typically taken to be $\approx 0.18$ ml g$^{-1}$ [12], enabling the cell mass to be determined by the above relation. This value of the RI increment is taken to reflect the average composition of cells, limiting it to an approximate value. It should also be noted that the cell mass should here be understood as the difference between the total mass enclosed within the volume of the cell and the mass of an equal volume of the surrounding cell medium. The cell mass relative to water, often denoted dry mass, is related to $m_{cell}$ as $m_{dry} = m_{cell} + \left(\frac{\mathrm{d}n}{\mathrm{d}c}\right)^{-1} V_{cell}(n_{med} - n_w)$, where $n_w$ denotes the RI of water. In the following, by "cell mass" we will mean the dry mass of the cells. This approach to relate RI to dry mass has been used previously to study cellular growth rate[14], quantifying mass

densities of intracellular structures[15], in addition to phenotyping and characterization of pathological cell types in a variety of diseases[2–5,16], among others.

However, since the phase shift is composed of the product of cell thickness and RI (Eq. (1)), an unambiguous and quantitative separation of these two parameters is non-trivial. That is, while the product can be determined with high accuracy, either cell thickness or RI needs to be assumed or independently determined in order to decipher the other parameter. Isolating these parameters can be accomplished in several ways: multiple illumination angles allows for tomographic holographic imaging[17], dual wavelength holography in combination with a highly dispersive medium[18], or by sequential exposure to two different media having different RI, $n_1$ and $n_2$[19]. The latter approach makes use of the fact that the cell volume $V_{cell}$ and cell RI $n_{cell}$ can be expressed in terms of the integrated phase shifts (introduced above) measured sequentially in the two media, $\Phi_1$ and $\Phi_2$, as

$$kV_{cell} = \frac{\Phi_1 - \Phi_2}{n_2 - n_1}, \tag{3}$$

$$n_{cell} - n_2 = \frac{\Phi_2}{kV_{cell}}, \tag{4}$$

which follows from evaluating Eq. (2) at two different medium RI. Thus, by controlled solution exchange around the sample, details of the cellular state can be elucidated with respect to cell mass, volume, and RI.

To quantify and correlate mass, volume and RI of individual yeast cells (*Saccharomyces cerevisiae*), we employ in this work a digital holographic microscope (DHM) in combination with mathematical modeling and a millifluidic chip, providing quantitative phase information with temporal control of the cellular environment. Furthermore, by monitoring cellular volume as a function of external osmolarity, we are able to parametrize a mathematical model for the osmotic response. This model allows us to deduce the intracellular osmolarity and the non-osmotic volume of individual cells (i.e. the volume that is unaffected by changes in the external osmolarity), and to correlate these parameters to cell size. Finally, we quantify and monitor the temporal evolution of mass and volume of individual cells in response to an increased external osmolarity. By relating the response in mass and volume to our model for osmotic response, we are able to quantify the rate of osmoregulatory accumulation of osmolytes, as well as characterizing the produced osmolytes, in terms of the product of their RI increment and molar mass at the single-cell level. Utilizing this dynamic response, we demonstrate the applicability of QPI to quantify and monitor a multitude of biophysical parameters at the single microbial cell level, with high spatiotemporal selectivity. In particular, the possibility to characterize compounds produced in response to stress, while temporally resolving their rate of production, contributes a unique feature, which promises wide spread applicability beyond the cellular system explored in this work.

## Results

**Characterization of cell RI, volume, and mass**. To characterize the cellular response to changes in external osmolarity, we first exploited our ability to continuously monitor the phase shift of light passing through individual cells, while exchanging the extracellular medium in a controlled manner. Utilizing baker's yeast as a model, we immobilized cells onto the floor of a millifluidic chip (see Methods for detailed protocol). These immobilized cells were sequentially exposed to two different media, having RI of 1.338 and 1.363, respectively (see Methods). In order to employ Eqs. (3) and (4) for the determination of cell thickness

and cell RI, the media were chosen to be iso-osmotic. The measured phase shift $\hat{\phi}$ (see Methods) is characterized by a step-wise change as the external RI changes, while the cell area remains approximately unaffected (see red and blue curve in Fig. 1a). This change in the phase enabled us to determine the cell RI and the cell volume $V_{cell}$ based on Eqs. (3) and (4). It was elucidated that cells have an average RI of $\bar{n}_{cell} = 1.411$ (Fig. 1b) and a 25–75% percentile range of 1.402–1.420 ($N = 131$). The cells have a median volume of $\bar{V}_{cell} = 25$ fl (Fig. 1c) but demonstrate a large variability, with 25–75% quantiles at 17–37 fl ($N = 131$).

One contributor to the data distribution is the noise level of the system. Using setup-determined values, we estimate that the error in the phase shift, integrated over a circular region of radius $r$, is given by $\delta\Phi \approx 0.07(\text{rad} \cdot \mu m) \cdot r$ The uncertainty in the cell volume, as a result of this noise, then becomes $\delta V \approx (2 \cdot \frac{0.07 \, \text{rad} \cdot \mu m}{k(n_2 - n_1)} \cdot r) \text{fl} \approx 0.6 \, \mu m^2 \cdot r$ (see Supplementary Note 3). For a typical yeast cell with a radius of 2 $\mu$m, this amounts to an uncertainty of $\delta V \approx 1.1$ fl, or ~4%. Note, that this uncertainty estimation assumes negligible light refraction at the cell interface (see Supplementary Note 2 for a detailed discussion on this assumption). In addition, in deriving this estimate we have taken into account only the noise level of the optical system and neglected uncertainties stemming from determination of other system parameters (refractive indices of the imaging solutions, pixel size of the system, among others). However, as these system parameters apply equally to all measured cells, the uncertainty in the cell-to-cell variability can be expected to depend largely on the noise level of the optical system.

Determining the cell volume also enables the sphericity of each cell to be assessed. By comparing the measured cell volumes to the volume of spheres with radius $r = \sqrt{A/\pi}$, where $A$ is the measured surface coverage area of the cells, we found that most cells were approximately spherical (inset of Fig. 1c). We also investigated the correlation between cell volume and cell dry mass (see Eq. 2 and the surrounding discussion). We found, not surprisingly, that cell mass scales linearly with cell volume over a wide range of measured cell volumes. However, mass and volume are not directly proportional; instead small cells appear to be more densely packed than large cells, which is in agreement with previous findings in bacteria[20]. We find that mass concentration is inversely related to cell volume, approaching 0.34 g ml⁻¹ for large cells (inset of Fig. 1d). This observation likely reflects the larger relative mass of essential cell components (e.g. cell wall, ribosomes, and other organelles) in small cells, which is consistent with the fraction of mitochondrial mass to total cell mass being higher in young buds compared to that of the mother cell[21].

**Determination of intracellular osmolarity and cytoplasmic volume.** As discussed above, the measured phase shift is determined by the mass concentration of the cells, which is primarily dominated by the concentration of heavy molecules and higher order structures. However, the cell size is determined by balancing the intracellular and extracellular osmotic pressures, primarily determined by the intracellular concentration of small molecules and ions.

As we will show in this section, the possibility to interrogate the mass and volume of individual cells independently enables us to discern the relative prevalence of heavy compounds (in terms of the excluded volume of such compounds) and light compounds (in terms of the molar number of osmolytes in the cytoplasm). In order to achieve this, we systematically explored the response in cell volume to changes in extracellular osmolarity. The cytoplasmic volume will respond to such osmotic stress by adjusting its water content. However, due to the excluded volume of

macromolecules and larger cellular structures, the entire cell volume will not be susceptible to osmotic changes, and hence, we hypothesized that the volumetric response to osmotic changes might allow the excluded volume of such structures to be directly determined. To test this hypothesis, the cells were sequentially exposed to a series of media with constant RI (1.352) but with increasing osmolarity $\Pi_{ext}$, to induce a variation in the cell volume. A typical trace of the phase shift ($\hat{\phi}$) and area ($A$) of an individual cell is shown in Fig. 2a (time points of solution change are indicated with arrows). As Eqs. (3) and (4) are strictly valid only under the assumption that the cell volume is not affected by the change in solutes, these equations cannot be directly employed in this case. Instead, we assume that the cells deform isotropically, such that the cell volume scales with surface occupying area as $V_{cell} \propto A^{\frac{3}{2}}$ (see Methods)[22]. Based upon this assumption we can therefore measure changes in cell volume using Eqs. (6)–(8) (see Methods).

Building upon our analysis that the cellular mass concentration depends on cell size (in the context of the results shown in Fig. 1d), we further attempted to estimate both the mass and volume of the cytoplasm and other cell components separately, as well as the number and average molar mass of the cytoplasmic osmolytes. Inspired by the model proposed by Klipp et al.[23] we developed a mathematical representation which allows the volumetric response of the cell to be related to relevant biophysical parameters, including the volume and mass of the cytoplasm. The cytosolic osmolarity $\Pi_{cyt}$ and the extracellular osmolarity $\Pi_{ext}$ are related via the turgor pressure $\Pi_t$ (i.e. the pressure acting on the cell wall by the plasma membrane) as $\Pi_{cyt} = \Pi_{ext} + \Pi_t$ (at steady state). Further, the cytosolic volume $V_{cyt}$ is assumed to be related to the cytosolic osmolarity as

$$V_{cyt} = \frac{N_{osm}}{\Pi_{cyt}} = \frac{N_{osm}}{\Pi_{ext} + \Pi_t}, \tag{5}$$

where $N_{osm}$ is the molar number of osmotically active components in the cytosol. Considering that the turgor pressure results from the volumetric elastic modulus of the encapsulating cell wall, this pressure is an increasing function of cell volume. Following ref. [23] we further assumed that the turgor pressure is related to the cell volume as $\Pi_t = \Pi_t^0(1 + \delta(V_{cell} - V_0))$ for $\delta(V_{cell} - V_0) > -1$, and $\Pi_t = 0$ otherwise. Here $\Pi_t^0$ and $V_0$ are the turgor pressure and cell volume at a reference osmolarity, and $\delta$ is a parameter which quantifies the rigidity of the cell wall. Consequently, at high extracellular osmolarities (for which $\delta(V_{cell} - V_0) < -1$), the turgor pressure $\Pi_t$ can be neglected compared to $\Pi_{ext}$. Then, in the limit of large osmotic pressure, the cell volume is expected to scale as $V_{cell} \sim V_p + \frac{N_{osm}}{\Pi_{ext}}$, where $V_p$ is the osmotically inactive part of the cell volume[24]. Based on the response in the high-osmolarity regime, this allows for an estimation of both the molar number of osmolytes $N_{osm}$ and the excluded volume $V_p$. Figure 2b shows the inferred volume (the inferred cytosolic osmolarity as the inset) of the cell from Fig. 2a, as a function of the external osmolarity. This is presented together with a best fit line to the model presented above, using $\delta$, $V_p$, and $N_{osm}$ as the free parameters.

By fitting this model to the single cell responses, we found that the osmotically susceptible volume is linearly increasing with total cell volume (Fig. 2c, $N = 27$ cells). This susceptible cell volume will henceforth be denoted as cytoplasmic volume. Since a considerable fraction of intracellular water is bound to various cellular structures[25], it is further reasonable to consider this as a measure for the cellular free water content, i.e. the water which is free to cross the cell membrane. It should be noted that this volume includes the volume of intracellular membrane-bound

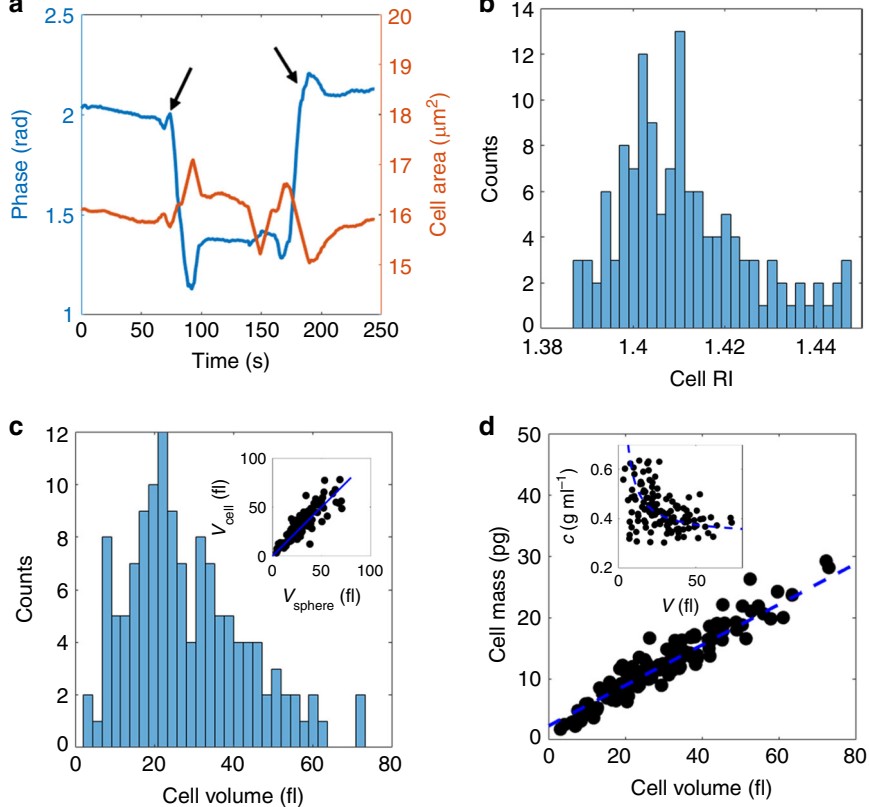

**Fig. 1** Determination of cell RI and volume. **a** The temporal response in the average phase shift ($\hat{\phi}$) (blue curve) and area (red curve) upon exposure of a yeast cell to two iso-osmotic solutes with different RI. The arrows indicate the time points of solute exchange. **b**, **c** The measured cell RIs and volumes based on the equations discussed in the main text. Inset to **c** shows the measured cell volume versus the expected volume for spheres of equal cross section to the imaged cells. Cells falling above the blue line are slightly prolate, cells falling below are slightly oblate. **d** Cell dry mass is strongly correlated with cell volume. The mass is fitted to an expression $m_{dry} = c_{cyt}V_{cyt} + c_pV_p$, with $V_{cell} = V_{cyt} + V_p$, where $V_{cyt}$ and $V_p$ represent the osmotically active and osmotically non-responsive volumes, determined by studying the scaling of volume with external osmolarity as detailed in the text. Inset: The mass concentration is inversely related to cell volume. The blue dashed line corresponds to the same fit as in the main figure

compartments which are also affected by the external osmolarity[26]. The cytoplasmic volume was found to scale linearly with cell volume (Fig. 2c) and the average cell (with volume greater than 15 fl) displayed a cytoplasmic volume ratio of 66% (at external osmolarity $\Pi_{ext} = 0.53$ Osm), consistent with previous findings[27]. It is interesting to note, that by extrapolating these results to smaller cells and buds, the cytoplasmic volume vanishes completely for cells smaller than ~5 fl (Fig. 2c), suggesting that the free water content is much lower in small cells. In fact, this result hints at an underlying model for cell composition and growth in budding yeast.

Young daughter cells appear to consist to a large extent of osmotically nonresponsive material, e.g. cell wall, organelles, and heavy molecules, such as proteins and RNA. Although being continuously synthesized, this osmotically nonresponsive cell fraction is slightly diluted upon cell growth, and occupies 30–35% of the cell volume in larger cells. In order to reconcile this with our findings demonstrating that the cell mass concentration is inversely related to cell size (Fig. 1d), we write the total cell mass as the sum of its contributions as $m_{cell} = c_{cyt}V_{cyt} + c_pV_p$, where $c_{cyt}$ and $c_p$ are the mass concentrations of the cytoplasmic and nonresponsive volumes. Inserting the dependency of the cytoplasmic volume on total cell volume found above (Fig. 2c), this expression is found to reproduce the dependency of mass on cell volume presented in Fig. 1d using $c_p = 0.79$ g ml$^{-1}$ and $c_{cyt} = 0.23$ g ml$^{-1}$ (see dashed line in inset to Fig. 1d), revealing that the nonresponsive volume accounts for ~64% of the total cell mass.

This model of partial intracellular dilution by addition of cytoplasmic volume with constant mass concentration, is corroborated by the observation that the number of osmolytes is found to be directly proportional to the cytoplasmic volume rather than total cell volume, suggesting that the intracellular osmolarity is volume-independent (Fig. 2d). Combining this with the quantification of cytoplasmic volume presented above, leads us to the determination that the average intracellular osmolarity is $\Pi_{cyt} = 0.60 \pm 0.04$ Osm (standard error of mean, $N = 27$) at $\Pi_{ext} = 0.53$ Osm, which agrees well with previously presented results[23]. Further, using $c_{cyt} = 0.23$ g ml$^{-1}$ as determined above, the average molar mass of cytoplasmic osmolytes was determined to be $M \approx 380 \pm 30$ g mol$^{-1}$ (standard error of mean, $N = 27$). It should also be noted that the value for the cytoplasmic mass concentration obtained above, relies on an accurate determination of the osmotically non-responsive volume, which can therefore only be employed as an estimate of the true cytoplasmic concentration.

**Monitoring and quantification of cellular uptake**. Finally, we exploit the capability to simultaneously monitor both cell mass and cell volume to investigate the long-time response of yeast cells to an osmotic shock. The aim of this measurement strategy is to elucidate if our method can be used to temporally resolve and quantify cellular uptake in terms of both molar number of accumulated osmolytes in addition to characterizing the nature of the osmolytes. It is known that yeast counter an increase in

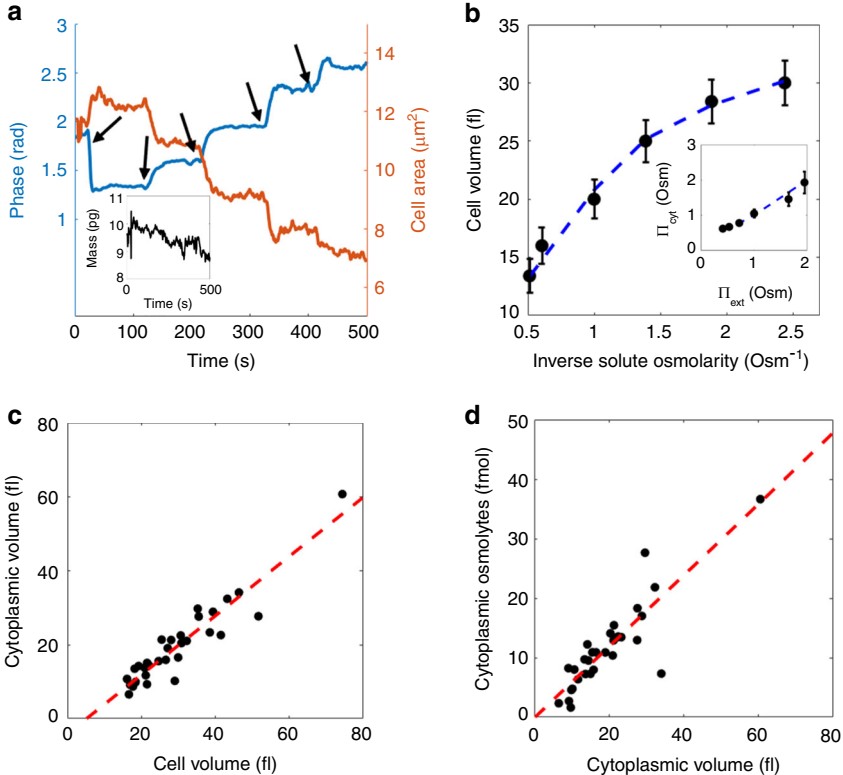

**Fig. 2** Response of yeast cells to brief (<10 min) osmotic shock. **a** Characteristic temporal response of phase shift $\hat{\phi}$ and area $A$ of a single cell during exposure to a series of solutes having identical RI but increasing osmolarity. The arrows indicate the time points of solute exchange. Inset shows the inferred cell mass, verifying that the observed changes in cell RI and volume are due to outflux of water only. **b** The volume of the cell analyzed in **a** as a function of inverse osmolarity. The dashed line shows a fit to the model described in the text. Error bars correspond to the uncertainty in the volume determination estimated in the main text. Inset: The cytosolic osmolarity as a function of external osmolarity. **c** The estimated cytoplasmic volume (related to the free water content) is linearly related to the cell volume (at external osmolarity $\Pi_{ext} = 0.53$ Osm). Small cells show smaller cytoplasmic volume (indicating less free water). **d** The number of intracellular osmolytes is directly proportional to cytoplasmic volume (at external osmolarity $\Pi_{ext} = 0.53$ Osm), suggesting a volume-independent cytoplasmic osmolarity

extracellular osmolarity by production and accumulation of osmolytes in the cytoplasm, primarily glycerol, which previously required determination by $^{13}C$-NMR in suspensions of lysed cells[28]. This accumulation of glycerol increases the intracellular osmolarity and enables water influx, thus restoring cell volume. By continuously monitoring the cell mass (or, more accurately, the product $\left(\frac{dn}{dc}\right)m_{dry}$, see Eq. (9)) and the cell volume, it is possible to determine the differential increase in cell mass with volume (which is here denoted $\left(\frac{dm}{dV}\right)$). This, we hypothesize, should allow the uptake and accumulation of osmoregulatory osmolytes to be distinguished from cellular water fluxes (for which $\left(\frac{dm}{dV}\right) = 0$) and from metabolic cell growth (for which $\left(\frac{dm}{dV}\right) = 0.34$ g ml$^{-1}$ as determined above). Further, assuming complete loss of turgor pressure of the cells immediately upon exposure to hyperosmotic shock, the change in cell volume can be directly related to the number of accumulated osmolytes via Eq. (10), allowing the initial rate of osmolyte production to be determined at the single cell level.

Relating this initial rate of osmolyte accumulation to the rate of change of cell mass, the product $K = \left(\frac{dn}{dc}\right)M$ can be determined (see discussion after Eq. (13)), where $M$ represents the average molar mass of the accumulated osmolytes. This physical characteristic is sometimes denoted the "molar refractive index increment" and represents the change in RI of a solution with the molarity of the solute. This differs significantly between biomolecules: $K \approx 0.011$ ml mol$^{-1}$ for glycerol (assuming $\left(\frac{dn}{dc}\right) = 0.115$ ml g$^{-1}$ and $M = 92$ g mol$^{-1}$), $K \approx 0.053$ ml mol$^{-1}$ for

trehalose, another biomolecule that can be produced during stress[29] (assuming $\left(\frac{dn}{dc}\right) = 0.155$ ml g$^{-1}$ and $M = 342$ g mol$^{-1}$) and $K \approx 1.9$ ml mol$^{-1}$ for proteins (assuming $\left(\frac{dn}{dc}\right) = 0.19$ ml g$^{-1}$ and $M = 10000$ g mol$^{-1}$). Consequently, estimating this quantity allows the accumulation of glycerol to be distinguished from the accumulation of other compounds.

To test this measurement strategy, the cells were imaged in regular growth medium (see Methods for details) for ~2 min before being exposed to a medium containing 0.5 M NaCl, 0.5 M sorbitol, and 2% glucose, with the latter component acting as a carbon source, allowing the cells to synthesize intracellular osmolytes. The phase maps of the cells were acquired at regular intervals for 1 h after exposure to the second medium. We observed two distinct phases of osmoadaptation: an initial rapid volume decrease, followed by a slow swelling of the cell. In line with our previous reasoning, the initial rapid volume decrease was assumed to be a result of water outflux. Since the RI of the cell medium was known (RI = 1.337 before osmotic upshift and RI = 1.355 after, as determined by an Abbe refractometer) this allowed us to isolate and monitor cell volume and cell mass. In Fig. 3, the traces of phase shift ($\hat{\phi}(t)$, Fig. 3b) and area ($A(t)$, Fig. 3c) are shown for the four individual cells in the colony highlighted in Fig. 3a. From these traces, the cell volume (Fig. 3d) and the change in cell dry mass ($m_{dry}$, shown in Fig. 3e) was quantified and monitored (assuming $\left(\frac{dn}{dc}\right) = 0.18$ ml g$^{-1}$). By investigating the cell mass as a function of cell volume, we found that the cell mass is strongly correlated with cell volume during

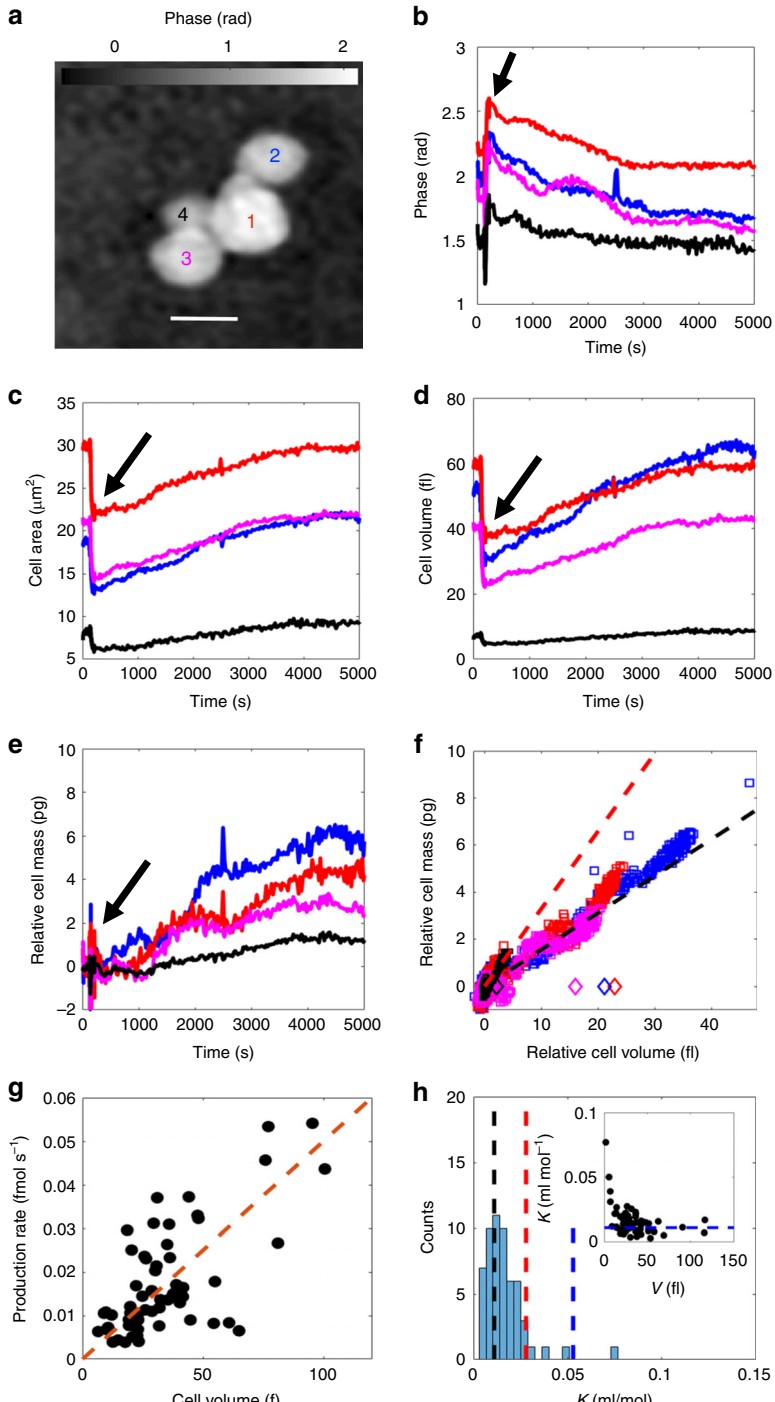

**Fig. 3** Characterization of cells under osmotic adaptation. **a–c** Monitored phase shifts ($\hat{\phi}$) (**b**) and projected cell areas (**c**) of the four cells shown in **a** during osmoadaptation (black arrow indicates time point of osmotic upshift). The line colors correspond to the cell numbers in **a**. Scale bar in **a** is 5 μm. **d**, **e** Inferred cell volume $V_{cell}(t)$ (**d**) and the relative cell mass $m_{dry}(t) - m_0$ (**e**), where $m_0$ denotes the cell mass prior to osmotic upshift. **f** Relative cell mass $m_{dry}(t) - m_0$ as a function of relative cell volume $V_{cell}(t) - V_m$, where $V_m$ denotes the minimum cell volume attained after initial water outflux, at different stages in the osmoadaptation for the cells shown in **a**. The initial relative cell masses and relative volumes are shown as open diamonds. Immediately upon exposure to a high osmolarity medium, the cells expel water and reduce in volume while retaining their mass, shifting horizontally in the figure. Thereafter the cells slowly expand while gaining in mass (colored squares). The black dashed line represents the expected cell mass increase for accumulation of glycerol. The red dashed line represents the expected mass increase with volume had the cell swelling been a result of metabolic cell growth. **g** The rate of osmolyte production increases will cell size ($V_m$) with a typical production rate of 0.5 mM s$^{-1}$ (red dashed line). **h** The estimated value of the parameter $K$ defined in the main text for $N = 57$ cells is consistent with accumulation of primarily glycerol ($K = 0.011$ ml mol$^{-1}$ shown as dashed black line). Also shown are the expected values of $K$ for glucose ($K = 0.028$ ml mol$^{-1}$ shown as red dashed line) and trehalose ($K = 0.053$ ml mol$^{-1}$ shown as blue dashed line) Inset: Small cells appear to accumulate compounds with slightly larger values of $K$. The blue dashed line corresponds to the value of $K$ for glycerol. The cell volume is here taken to mean the minimum cell volume after osmotic upshift ($V_m$)

the adaptation process, suggesting a continuous uptake and accumulation of biomaterial (Fig. 3f), and also that the differential cell mass increase is consistent neither with metabolic cell growth (dashed red line in Fig. 3f) nor with redistribution of cellular water (for which $\left(\frac{dm}{dV}\right) = 0$ as discussed above).

As a control, we exposed cells to a phosphate buffer containing 0.5 M NaCl and 0.5 M sorbitol but no glucose. In this case, no cell swelling was observed in any of the investigated cells ($N = 9$), suggesting that the accumulation of osmolytes requires uptake and subsequent metabolic breakdown[23] of an external carbon source (e.g. glucose).

Further, taking the approach outlined above, we are able to quantify the rate of osmolyte production at the single cell level (see Methods). We find that the production rate generally increases with cell size (Fig. 3g) with a typical production rate of ~0.5 mM s$^{-1}$. By further relating the production rate to the rate of cell mass increase, the parameter $K = \left(\frac{dn}{dc}\right)M$ was determined at the single cell level (see Methods) (Fig. 3h). We found that smaller cells typically accumulate heavier components (signified by a larger value of $K$) (inset in Fig. 3h), which suggests that cell growth and osmoadaptation may occur simultaneously. Cells with volume >30 fl show values of $K$ in the range 0.009–0.017 ml mol$^{-1}$. Since this range is consistent with glycerol (having $K \approx 0.011$ ml mol$^{-1}$) but well separated from most other biomolecules (see above), this strongly suggests that glycerol production is responsible for the observed cell swelling.

## Discussion

We have demonstrated that QPI in combination with local environmental control can be used to extract a number of previously unattainable biophysical cell parameters, such as intracellular osmolarity and cytosolic volume, at the single cell level. In addition, by correlating temporal changes in cell mass and cell volume, we monitor and quantify cellular uptake in terms of the rate of osmolyte accumulation, while simultaneously determining the product of the RI increment and the molar mass of the accumulated compounds. This allows the accumulation of glycerol to be distinguished from other higher molecular weight compounds. The method relies on an initial determination of cell volume and RI by exposing the cell to two solutes with different RI. Thereafter the volume is monitored by assuming that the cell volume scales as $A^{\frac{3}{2}}$, where $A$ is the area that the cell occupies in the microscopy image. Under this assumption the cell RI and volume can be inferred in an arbitrary environment, as long as the RI of the environment is known. We find that, on an ensemble average, cell dry mass and cell volume are linearly related over a wide range of cell sizes. However, we find that the cytosol of small cells (volume $V < 20$ fl) are more densely packed than large cells. We attribute this to a larger relative fraction of heavy cell components present in smaller cells.

The prevalence of such structures cannot be assessed by studying the phase shift in steady state. Instead, we compare the response in cell volume to hyperosmotic perturbations in the extracellular milieu to a mathematical model for the response, parametrized by the cytoplasmic osmolarity, the osmotically nonresponsive volume and the volumetric elasticity of the cell wall. In this way, we were able to estimate the intracellular osmolarity, as well as the fraction of osmotically active volume. This fraction was determined to be on average 66% of the total cell volume (corresponding to a nonresponsive volume of 34% of total cell volume). Further, the estimated mass of the passive volume (~64% of total cell mass) is consistent with the expected values for the yeast cell wall (15–30% of total cell mass)[30] together with the combined mass of lipids, protein, and RNA (~50% of total cell mass)[31].

The estimation of the cytosolic osmolarity relies on the assumption that the pressure exerted by the cytoplasm on the cell membrane is purely osmotic, i.e. that the exerted pressure is proportional to the molar concentration of impermeable solutes. However, this is strictly valid only when intramolecular interactions can be neglected. It is well known that the cytoplasm behaves as a viscoelastic liquid, with material properties that depend on cell state and environmental stress[6,7]. Thus, the viscoelastic nature of the cytoplasm will likely also influence the temporal response to an osmotic shock, and hence, by temporally resolving the initial stage of water outflux this method could in principle also be used to assess the mechanical properties of the cytoplasm under non-equilibrium situations.

Under the assumption that the pressure exerted by the cytoplasm on the cell membrane is of purely osmotic origin, we were also able to quantify changes in the intracellular environment in terms of the average molar mass and number of cytoplasmic osmolytes, by inducing uptake and accumulation of glycerol through a prolonged exposure to osmotic stress. While this method can distinguish between breakdown of cytoplasmic compounds and cellular uptake, it does not uniquely identify the compounds that pass the cell membrane. For instance, it is not presently possible to distinguish between direct uptake of glycerol and uptake of higher order saccharides which are subsequently decomposed into glycerol. Nonetheless, we believe that the possibility of label-free determination of both rate of uptake, as well as a molecular identifier of compounds accumulated inside or excreted from lipid-membrane bound compartments, will find applications far beyond the present study. In particular, the spatiotemporal resolution of holographic microscopy may enable mass transport to and from intracellular compartments to be studied with a similar approach. Label-free spatio-temporal imaging and quantification of cellular and intracellular mass transport in this manner may provide a crucial complement to existing label-based and label-free approaches.

## Methods

**Optical setup**. A sketch of the setup employed in this study is shown in Supplementary Figure 1. A 633 nm HeNe-laser (Newport) is split into two light paths, one passing through the sample and one which does not. The two beams are recombined at a slight offset angle, and the resulting interference pattern is recorded by a CCD-camera (AlliedVision, ProSilica GX1920). The offset angle splits the interference pattern into three separated peaks in the Fourier space. Analyzing these peaks allows a quantitative phase map of the sample to be constructed, as described below.

**Image analysis**. The interference patterns, or holograms, were analyzed using the software MATLAB (Mathworks Inc.) to extract the amplitude and phase maps using standard methods[32]. In brief, due to the off-axis configuration of our setup, the Fourier transform of the interference pattern contains two off-center peaks, which describe the object field multiplied by a plane wave described by $\exp(\pm i\mathbf{k}_p \cdot \mathbf{x})$. Here, $\mathbf{k}_p$ represents the projection of the wave-vector of the reference beam onto the imaging plane (camera). In order to isolate the object field, we numerically shifted one of the off-center peaks to the center of the Fourier spectrum and applied a low-pass filter (see Supplementary Note 1 for further details). The magnitude and phase of the resulting field correspond to the amplitude and phase of the optical field recorded by the camera. The obtained phase map is slightly distorted due to optical aberration in the beam line. This was corrected in the post-processing step by fitting the phase map to a parabola, which was subsequently subtracted from the phase map. Numerical autofocus was implemented using the focus criterion described by Sun et al.[33] and propagating the obtained field using the angular spectrum method[32]. In order to avoid ambiguities in the data analysis stemming from threshold settings and overlapping cells, the phase response of the cells was fitted to a circularly symmetric function $\Psi = \Psi_0 + \hat{\phi} \tanh\left(\frac{\rho_0 - \rho}{\sigma}\right)$, where $\Psi_0$ denotes the background phase, $\rho$ is a radial coordinate defined from the center of the cell and $\sigma$ represents the fall-off of the phase shift, primarily related to the spatial resolution of the microscope. Assuming that $\sigma \ll \rho_0$, the parameters $\rho_0$ and $\hat{\phi}$ are related to the cell radius $r$ and the integrated phase shift $\Phi$, as $r \approx \frac{7}{6}\rho_0$ and $\Phi \approx \pi\hat{\phi}\rho_0^2$ (see Supplementary Note 2). The function ($\Psi$) was found from numerical simulations to correctly reproduce the phase response of numerical model cells in environments of varying RI, and thus provided an objective method to estimate the cell size. This enabled the surface occupying areas, $A = \pi r^2$, and phase shift $\hat{\phi}$, of individual cells to be tracked as a function of time.

**Cell handling**. Yeast cells were cultured in a synthetic complete (SC) medium (Sigma Aldrich) in the presence of 2% glucose overnight. Cells were then diluted and regrown for ~3 h prior to imaging to ensure that the cells were imaged during their log-phase. The imaging was performed inside either homemade or commercial (Ibidi sticky-slides VI) millifluidic chips which were treated with Concanavalin A (Sigma Aldrich, 0.1% w/v) to ensure cell immobilization.

**Imaging solutions**. To separate the contributions of cell thickness from cell RI to the measured phase shift under iso-osmotic conditions, the cells were initially imaged in a 200 mM phosphate buffer with 2% (w/v) glucose (RI = 1.338, measured by an Abbe refractometer). Next, cells were briefly (~30 s) exposed to an iso-osmotic solution consisting of 20% (w/v) sucrose (Sigma Aldrich) (RI = 1.363).

To study the scaling of cell volume with osmolarity, the cells where initially imaged in a 200 mM phosphate buffer with 2% (w/v) glucose as above (calculated $\Pi_{\text{ext}} = 0.53$ Osm), whereafter the cells were exposed to a series of NaCl–sucrose solutions (14% (w/v) sucrose + 0% NaCl, 13% sucrose + 1% NaCl, 12% sucrose + 2% NaCl, 9% sucrose + 4% NaCl, and 8% sucrose + 5% NaCl) with increasing osmolarities (calculated $\Pi_{\text{ext}} = 0.41, 0.72, 1.0, 1.65, 1.95$ Osm) but a fixed RI (RI = 1.352, measured by an Abbe refractometer).

To quantify the osmoregulatory response, the cells were initially imaged in SC medium in the presence of 2% glucose (RI = 1.337, measured by an Abbe refractometer, $\Pi_{\text{ext}} = 0.29$ Osm[6]). Thereafter the cells were exposed to SC medium supplemented with 0.5 M NaCl, 0.5 M sorbitol, and 2% glucose (RI = 1.355, measured by an Abbe refractometer, calculated $\Pi_{\text{ext}} = 1.8$ Osm)

**Numerical simulations**. To estimate the effect of refraction and to aid in objectively quantifying the experimentally obtained phase maps, we performed numerical simulations based on the beam propagation method. The beam propagation method utilizes the angular spectrum method with an additional local phase correction to compensate for inhomogeneities in the RI of the system (assuming unit transmission)[34]. The cells were defined as spherical structures containing a core of uniform RI (representing the cytoplasm) and a thin shell of higher RI (representing the cell wall). This enabled us to define a symmetric basis function $\Psi$ which was found to accurately reproduce the phase response of the simulated cells.

**Quantification of cell uptake and monitoring cell mass and volume**. In order to quantify processes in which the cell mass and/or volume changes dynamically, we assume that the cell sphericity $\epsilon = V_{\text{cell}}/V_{\text{sphere}}$ is conserved, where $V_{\text{sphere}}$ is the volume of a sphere with the same occupying area as the cell. This assumption implies that the cell volume scales with the surface occupying area (which is directly determined from the microscopy image) as $V_{\text{cell}} \propto A^{\frac{3}{2}}$, which allows the cell volume to be monitored once the cell volume at one time-point is determined. Assuming that the cell dry mass immediately before and after solute change is conserved, i.e. that at short time-scales the cell respond to changes in solute osmolarity by water redistribution, the ratio of the volumes $\frac{V_{\text{cell,1}}}{V_{\text{cell,2}}}$ before and after solute exchange can be directly determined from the microscopy image by computing the ratio of the surface occupying area of the cell before and after the water outflux, $A_1/A_2$, as

$$\frac{V_{\text{cell,1}}}{V_{\text{cell,2}}} = \left(\frac{A_1}{A_2}\right)^{\frac{3}{2}} \equiv \delta^{-1}. \qquad (6)$$

This allows the initial cell volume to be determined as

$$kV_{\text{cell,1}} = [(1-\delta)n_{\text{w}} + \delta \cdot n_2 - n_1]^{-1}(\Phi_1 - \Phi_2), \qquad (7)$$

and the volume can be monitored as

$$V_{\text{cell}}(t) = V_{\text{cell,1}}\left(\frac{A(t)}{A_1}\right)^{\frac{3}{2}}. \qquad (8)$$

The cell dry mass is then quantified as

$$m_{\text{dry}}(t) = \left(\frac{dn}{dc}\right)^{-1}\left(k^{-1}\Phi(t) + V_{\text{cell}}(t)(n_2 - n_{\text{w}})\right). \qquad (9)$$

Assuming loss of turgor pressure, a change in the number of intracellular osmolytes $\delta N$ induces a change in the cell volume $\delta V$ according to

$$\delta N = \Pi_{\text{ext}}\delta V. \qquad (10)$$

In order to estimate the production rate $\lambda$ of osmolytes, an exponential approach, $\delta V(t) \sim \delta V_{\text{tot}}(1 - \exp[-\mu(t - t_0)])$, was fitted to the change in cell volume during the adaptation, where $\delta V_{\text{tot}}$ denotes the total change in cell volume

during the adaptation, $t_0$ indicates the time point at which cell swelling begins and $\mu^{-1}$ sets the time scale for cell swelling. The rate of osmolyte production was then defined as

$$\lambda \equiv \Pi_{\text{ext}}\delta V_{\text{tot}}\mu. \qquad (11)$$

Assuming that the osmolytes are produced by uptake and subsequent breakdown of extracellular compounds, the cell mass (relative to water) is related to the number of produced osmolytes via

$$m_{\text{dry}} = m_0 + M\delta N, \qquad (12)$$

where $m_0$ is the initial cell mass and $M$ is the average molar mass of the produced osmolytes. Taking the time-derivative of this equation we find

$$\frac{\partial m_{\text{dry}}}{\partial t} = M\frac{\partial \delta N}{\partial t}. \qquad (13)$$

At the onset of osmoregulation, we have that $\frac{\partial \delta N}{\partial t} = \lambda = \Pi_{\text{ext}}\delta V_{\text{tot}}\mu$ from Eq. (11) which is solely determined from the volume response. Thus, the only remaining parameter determining the mass response of the cell during osmoregulation is the molar mass $M$ of the accumulated osmolytes. However, since the cell dry mass is known only up to a multiplicative constant $\left(\frac{dn}{dc}\right)$ (see Eq. (9)), the experimentally determined parameter is not the molar mass of the accumulated compounds but rather the product $K \equiv \left(\frac{dn}{dc}\right)M$.

**Data handling**. In analyzing the data, cells that were inconsistently segmented and/or displaying estimated values of RI and/or volume not falling within an expected initial range ($0.03 < n_{\text{cell}} - n_{\text{w}} < 0.14$, $0 < V_{\text{cell}} < 150$ fl) were disregarded from the analysis. All experiments were performed in at least duplicate. Sample sizes were chosen to ensure that a broad distribution of cell sizes was represented in the data, thus allowing for correlation of the estimated parameters with cell size.

**Reporting summary**. Further information on experimental design is available in the Nature Research Reporting Summary linked to this article.

## Data availability
All data is available from the authors on request.

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

## Acknowledgements

D.M. acknowledges the Simon Alberti Group (MPI-CBG, Dresden) for providing cells, Beidong Liu (Gothenburg University) for storage of cells, and Niek Welkenhuysen (Gothenburg University) for cell medium preparation. The authors acknowledge support from the Swedish research council, grant number 2018-04900 and the Nanotechnology Area of Advance at Chalmers University of Technology.

## Author contributions

D.M., F.H., and G.D.M.J. designed the study. D.M. and G.D.M.J. designed and constructed the experimental setup. E.O. performed numerical simulations. All authors contributed in data analysis and interpretation, and in writing the manuscript.

## Additional information

**Competing interests:** The authors declare no competing interests.

