## [Peer Review File · Nature Communications]

Reviewers' comments:

Reviewer #1 (Remarks to the Author):

Reviewer's comments about the manuscript "Label-free, spatio-temporal monitoring of cytosolic mass, osmolarity and volume, in living cells by Daniel Midtvedt et al..

Authors propose a work in which they use digital holographic microscope in combination with a microfluidic chip, for investigating osmoregulatory response of yeast at single-cell scale. Thus they show changes in both intracellular osmolarity and macromolecular concentration when local environment conditions change.

Major comments:

1) The material about Fig.1C and Fig 1D is redundant as it is a trivial and very well known way to processing digital holograms and thus that plot are not significant for the readers and should be eliminated by the manuscript. Nonetheless, as anybody expert in DH knows the configuration with straight vertical fringes is not the most optimized configuration for off-axis DH.

2) page 6: the expression "step-wise drop" can induce confusion. Why to use the word "drop" ?? Please change.

3) Page 6: "...external RI changes, while the cell area remains unaffected (figure 2A)." it is not clear by the text and also by looking to the plots in Fig.2A what such plots are.

What means: "temporal response in phase shift (blue curve)"? The phase shift is related to a single point? Or to a line? Or to an area? .. an integration below a curve or surface??

What means: "...and area (red curve)" same questions above.

.. even because "phase shift" has not defined in unique way in the rest of the manuscript. Auhtrios at page 2 wrote: "the integrated phase shift over the area of the specimen".. ???

PLEASE CLARIFY!! ...and furnish operative definitions for the quantities plotted in FIG.2

MAIN CRITICISM can be summarized by reading the conclusions:

4) Authors conclude: "Our results thus pave the way for performing label-free quantification and monitoring of the intracellular mass distribution, osmolarity and cytosolic structure, providing a novel complementary approach to study a wide range of cellular processes".

Essentially all the measurements and all aspects of the presented method are not NEW. Capabilities of the DH are very well known. Thus, by the point of view of the method the paper cannot be claimed as an ORIGINAL work. On the other side, the work by the biological point of view seems interesting. However the authors emphasize the METHOD rather than a NEW BIOLOGICAL INSIGHTS OR significant RESULTS have been achieved in their work. Consequently this reviewer cannot be favorable for accepting this work unless the authors prove that a completely new biological step forward has been obtained by a well-known methods (i.e. DH).

In summary, in consideration of the above criticism this reviewer considers that this manuscript cannot be considered for publication in the journal but it is rather more suitable for Scientific Report journal instead.

Reviewer #2 (Remarks to the Author):

The manuscript presents data supporting label-free continuous monitoring of key cellular parameters such as volume and osmolarity. This is possible since the authors take the cellular response to the environmental perturbation into account. The presented data and analysis generally confirm previous model predictions of the temporal pattern of osmoregulation in yeast cells.

MAJOR COMMENTS

1. Logical flow

There are shortcomings in the logical flow of the manuscript. Key equations are hidden in the text, not comprehensively motivated, and sometimes occur in non-optimal order. Besides, notation is sometimes ambiguous. There are also several derivations in the Results section that might suit better in the Materials and methods section. In addition, there are numerous minor errors (see list below). Taken together, the manuscript is hard to read. A couple of specific suggestions regarding the logical flow:

Define and clearly motivate the phase-shift (ϕ) equation (Line 40) as a stand-alone Equation and not hidden in the text. After the definition of ϕ (Line 40), facilitate for the reader by unambiguously defining Φ by an equation and in layman terms. For the latter, why not use the explanation from Zangle and Teitell (2014): "integration is performed across the entire cell area, A"?

Next, define and clearly motivate the cell-mass equation (now on Line 153) in the Introduction (as a stand-alone Equation and not hidden in the text). This relationship is fundamental to appreciate the manuscript. In the current manuscript, the relationship between relative RI and cell mass on line 41 is ambiguous. Clearly explain what the RI is relative with respect to. Furthermore, is this an empirical relationship, and if so, what are the assumptions behind it? In addition, is it assumed that the environment is constant?

Eq 1-2. To appreciate these equations the reader must be explicitly informed about the definition of relative RI, Δn . While the Δ indicates a difference, hence $\Delta n = n_{\text{cell}} - n_{\text{environment}}$, the term "relative" may be interpreted as "in proportion to", hence $\Delta n = n_{\text{cell}} / n_{\text{environment}}$. From the equations, one sees that the authors are using the first interpretation. To avoid confusion, consider using another term than "relative RI" in the text (RI difference?).

Notation: Δn was defined as relative RI to the environment in the Introduction. On line 189, the same notation seems to denote how much two solutes differ in RI.

Line 195: Clarify how the reasoning of cell sphericity influences the cell-mass assumption.

Line 204-205: I fail to reproduce the derivation of the $z(t)$ formula.

$\epsilon = V_{\text{sphere}} / V_{\text{cell}}$ (from Line 189)

$V_{\text{sphere}} = \frac{4}{3} \times \pi \times r^3 = \frac{4}{3} \times \pi \times (\sqrt{A/\pi})^3$ (from Line 190)

hence, $V_{\text{cell}} = \frac{4}{3} \times A^{1.5} \times \pi^{-0.5} \times \epsilon^{-1}$

$z = V_{\text{cell}} / A$ (from Line 203)

hence $z = \frac{4}{3} \times A^{0.5} \times \pi^{-0.5} \times \epsilon^{-1}$ (which is not identical to the presented formula in the manuscript)

In summary, lay out the central equations in a logical order, clearly motivate each equation, and list potential assumptions underlying each equation.

2. Uncertainty estimations

The detailed uncertainty calculation for volume determination is questionable given that other key assumptions (e.g. the assumption of no impact of light refraction at the cell-medium interface on Line 40-41, and subsequent assumption of the RI increment of 0.18 mL/g on Line 155) were not quantified. Many aspects of the mathematical models of osmoregulation also include assumptions

that are not quantified.

3. The pressure model

Line 219. The pressure relationship only holds at steady state.

Line 240-241. The estimate of osmotically susceptible volume should be compared to relevant literature. For example, a comprehensive discussion of the non-osmotic volume, literature references, and a model-based estimation from data are given in <https://www.ncbi.nlm.nih.gov/pubmed/16922683>. The estimate is 0.37, and hence the osmotically susceptible volume is 63% (which, notably, is very similar to the estimate (66%) in the present manuscript).

Figure 4A. It would be of interest to see time-resolved data for cell mass (volume is already shown in the inset). Data of this kind are of high interest to the modelling community to revise or refine current state-of-the-art models of yeast osmoregulation. The authors should add all relevant meta-information about the experiment (such as type, level and time-points of the added osmotic stress, and are recommended to add raw data to the supplementary material.

4. Perspective

How do the findings of a linear relationship between cell dry mass and cell volume (for medium to large cells) and of higher cytosol density in cells with volume smaller than 20 fL relate to previous data in the literature?

MINOR COMMENTS

Consistently insert a space between number and unit. There are several instances to correct, e.g. on lines 79, 87, 92, 125, 127, 129, 148, 149, 170, 298, 310, 312, and 328.

Line 40: effects...is -> effects...are

Line 45: is composed of -> depends on

Lines 86, 134 and 335: The latin name for Baker's yeast was given in the Introduction, no need to repeat.

Line 100: corresponds -> correspond

Line 111: what is meant by "numerical model cells"?

Line 116-117: Use the abbreviation BPM (just introduced).

Line 136: Most of this already described in Materials and methods.

Line 153: What is "obtained as": the correlation, the volume or the mass?

Line 174: space between "intra-" and "and".

Line 175: unclear what "which is" refers to.

Lines 175-179: This sentence is too long, very hard to interpret, and (likely) grammatically incorrect.

Line 215: Hohmann et al. -> Klipp et al.

Line 268: contnent -> content

Line 375: determine -> to determine

Figure 1A: Components in the figure are undefined.

Figure 2A: The time-points of solute exchange should be indicated.

Figure 2C: What is meant by the red dashed line in the inset?

Figure 2 legend: Inset to D not defined. Cannot see a red dashed line or a blue dotted line in Figure D.

Figure 3A: The time-points of solute exchange should be indicated. Which solutes were included?

Figure 3 legend. Define "brief".

Figure 4 legend. is shown -> are shown (Line 321)

Reviewer #3 (Remarks to the Author):

The authors present a label-free imaging technique (digital holography microscopy) to visualize osmoregulatory response of individual yeast cells due to external perturbation of the local environment.

The work is highly novel and would be of great interest to many researchers in the field of osmoregulation, mainly due to the low cost of the imaging system and low technical difficulties of rebuilding a similar system.

Even though the manuscript can be published as is, the image analysis part would benefit from a more detailed explanation why a shift to the center of the Fourier spectrum is necessary. To my knowledge this is not a maneuver that is commonly used, even though every pixel is treated equal in the process.

My recommendation is to accept the manuscript for publication.

Response to Reviewers Comments

Label-free, spatio-temporal monitoring of cytosolic mass, osmolarity and volume, in living cells

Daniel Midtvedt, Erik Olsén, Fredrik Höök & Gavin D. M. Jeffries

We kindly thank the reviewers as well as the editor for their evaluation of this manuscript. Please find below the itemized responses to the comments and inserted manuscript statements. In conjunction with this document, please find a revised manuscript and supplementary information, containing all of the changes, and a separate document where all changes to the manuscript have been highlighted for ease of identification.

Referee #1:

Reviewer's comments about the manuscript "Label-free, spatio-temporal monitoring of cytosolic mass, osmolarity and volume, in living cells by Daniel Midtvedt et al..

Authors propose a work in which they use digital holographic microscope in combination with a millifluidic chip, for investigating osmoregulatory response of yeast at single-cell scale. Thus they show changes in both intracellular osmolarity and macromolecular concentration when local environment conditions change.

Major comments:

1) The material about Fig.1C and Fig 1D is redundant as it a trivial and very well known way to processing digital holograms and thus that plot are not significant for the readers and should be eliminated by the manuscript. Nonetheless, as anybody expert in DH knows the configuration with straight vertical fringes is not the most optimized configuration for off-axis DH.

While we agree that the material presented in figure 1 is well known for experts in the field of digital holography, we believe that the readers of Nature Communications, which is a multidisciplinary journal, are aided by a step-by-step description of the process of constructing a quantitative phase map. However, as this illustrative process does not provide any new information to the manuscript as a whole, we have chosen to move the figure to the supplementary information.

2) page 6: the expression "step-wise drop" can induce confusion. Why to use the word "drop" ?? Please change.

We agree that this wording might be confusing. We have changed it to "step-wise change" instead.

3) Page 6: "...external RI changes, while the cell area remains unaffected (figure 2A)." it is not clear by the text and also by looking to the plots in Fig.2A what such plots are. What means: "temporal response in phase shift (blue curve)"? The phase shift is related to a single point? Or to a line? Or to an area? An integration below a curve or surface?? What means: "...and area (red curve)" same questions above. Even because "phase shift" has not been defined in unique way in the rest of the manuscript. Authors at page 2 wrote: "the integrated phase shift over the area of the specimen".???

PLEASE CLARIFY!! ...and furnish operative definitions for the quantities plotted in FIG.2

We apologize for not having been clear on these points. Clarifications have been made to the manuscript, centered around figure 2A, that by "phase shift" we mean the parameter $\hat{\phi}$, as now clearly defined in Materials and Methods, and by "cell area" we mean the area that is occupied by the cell in the microscope image.

MAIN CRITICISM can be summarized by reading the conclusions:

4) Authors conclude: "Our results thus pave the way for performing label-free quantification and monitoring of the intracellular mass distribution, osmolarity and cytosolic structure, providing a novel complementary approach to study a wide range of cellular processes".

Essentially all the measurements and all aspects of the presented method are not NEW. Capabilities of the DH are very well known. Thus, by the point of view of the method the paper cannot be claimed as an ORIGINAL work. On the other side, the work by the biological point of view seems interesting. However the authors emphasize the METHOD rather than a NEW BIOLOGICAL INSIGHTS OR significant RESULTS have been achieved in their work. Consequently this reviewer cannot be favorable for accepting this work unless the authors proof that a completely new biological step forward has been obtained by a well-known method (i.e. DH).

In summary, in consideration of the above criticism this reviewer considers that this manuscript cannot be considered for publication in the journal but it is rather more suitable for Scientific Report journal instead.

We see the point of this opinion, and following the suggestion of the referee and the editor we have revised the manuscript in order to put more emphasis on the novel biological aspects of this work, although the combined methodological and analytic approach in our view do offer novel insights. To meet this concern, we have

- a. Added a paragraph in the introduction regarding the importance of characterizing dynamical intracellular processes at the single cell level. We replaced the sentence "Elucidation of cellular content by disentangling complex phase information opens up avenues for continuous monitoring of cytosolic composition with high degree of spatiotemporal selectivity." by "In this way, we demonstrate the applicability of quantitative phase imaging to quantify and monitor a multitude of biophysical parameters at the single microbial cell level with high spatiotemporal selectivity. In particular, the possibility to characterize compounds produced in response to stress and temporally resolve the rate of production, contributes a unique feature that promises wide spread use of the approach beyond the cellular system explored in this work." to more adequately highlight that the novelty of the paper lies not within

- the imaging technique itself (quantitative phase imaging) but rather at the means by which biophysical parameters were extracted (cytoplasmic volume, intracellular osmolarity, production rate of intracellular osmolytes and their characteristics etc.)
- b. Expanded the analysis of the data on osmoadaptation in Figure 4 (now figure 3), to show that the molar mass (or, more accurately, the product of $\left(\frac{dn}{dc}\right)$ and molar mass) of the produced osmolytes during adaptation depends inversely on cell size and that the production rate of osmolytes increases with cell size. This type of information was previously unattainable, as previous measurements on osmoadaptation was performed on an ensemble level only and, as pointed out by referee #2, single cell data of this type is highly valuable to the biophysical community to refine existing models on osmoadaptation.

Referee #2:

The manuscript presents data supporting label-free continuous monitoring of key cellular parameters such as volume and osmolarity. This is possible since the authors take the cellular response to the environmental perturbation into account. The presented data and analysis generally confirm previous model predictions of the temporal pattern of osmoregulation in yeast cells.

MAJOR COMMENTS

1. Logical flow

There are shortcomings in the logical flow of the manuscript. Key equations are hidden in the text, not comprehensively motivated, and sometimes occur in non-optimal order. Besides, notation is sometimes ambiguous. There are also several derivations in the Results section that might suit better in the Materials and methods section. In addition, there are numerous minor errors (see list below). Taken together, the manuscript is hard to read. A couple of specific suggestions regarding the logical flow:

Define and clearly motivate the phase-shift (φ) equation (Line 40) as a stand-alone Equation and not hidden in the text. After the definition of φ (Line 40), facilitate for the reader by unambiguously defining Φ by an equation and in layman terms. For the latter, why not use the explanation from Zangle and Teitell (2014): "integration is performed across the entire cell area, A"?

Next, define and clearly motivate the cell-mass equation (now on Line 153) in the Introduction (as a stand-alone Equation and not hidden in the text). This relationship is fundamental to appreciate the manuscript. In the current manuscript, the relationship between relative RI and cell mass on line 41 is ambiguous. Clearly explain what the RI is relative with respect to. Furthermore, is this an empirical relationship, and if so, what are the assumptions behind it? In addition, is it assumed that the environment is constant?

Eq 1-2. To appreciate these equations the reader must be explicitly informed about the definition of relative RI, Δn . While the Δ indicates a difference, hence $\Delta n = n_{\text{cell}} - n_{\text{environment}}$, the term "relative" may be interpreted as "in proportion to", hence $\Delta n = n_{\text{cell}} / n_{\text{environment}}$. From the equations, one sees that the authors are using the first interpretation. To avoid confusion, consider using another term than "relative RI" in the text (RI difference?).

Notation: Δn was defined as relative RI to the environment in the Introduction. On line 189, the same notation seems to denote how much two solutes differ in RI.

Line 195: Clarify how the reasoning of cell sphericity influences the cell-mass assumption.

Line 204-205: I fail to reproduce the derivation of the $z(t)$ formula.

$\epsilon = V_{\text{sphere}} / V_{\text{cell}}$ (from Line 189)

$V_{\text{sphere}} = 4/3 \times \pi \times r^3 = 4/3 \times \pi \times (\sqrt{A/\pi})^3$ (from Line 190)

hence, $V_{\text{cell}} = 4/3 \times A^{1.5} \times \pi^{-0.5} \times \epsilon^{-1}$

$z = V_{\text{cell}} / A$ (from Line 203)

hence $z = 4/3 \times A^{0.5} \times \pi^{-0.5} \times \epsilon^{-1}$ (which is not identical to the presented formula in the manuscript)

In summary, lay out the central equations in a logical order, clearly motivate each equation, and list potential assumptions underlying each equation.

We gratefully thank the referee for very relevant suggestions on how to improve the readability of the paper. Along the lines of his/her suggestions, we have:

- a. Introduced the phase contrast ϕ as a standalone equation
- b. Defined the integrated phase shift Φ as a standalone equation in the introduction, and clearly stated the relation of this quantity to cell mass.
- c. Clearly stated that Δn corresponds to the *difference* between cell RI and medium RI, and use this definition of Δn consistently throughout the manuscript.
- d. Clarified how assuming a constant cell sphericity allows us to quantify and monitor the temporal evolution of the cell volume. In brief, the equations relating the measured phase in two solutes to cell volume and cell RI implicitly assumes that volume and RI are unaffected by the change of solutes. Since this is not the case when the osmolarity of the solute is changed, we assume that the cell volume is related to the surface occupying area as $V_{\text{cell}} \propto A^{\frac{3}{2}}$. Relating these parameters allows to quantify the cell volume before and after water outflux. This finally gives us the opportunity to quantify processes which involve mass exchange with the environment (such as osmoadaptation). Changes have been made to the introduction of the manuscript to in a straight and hopefully more logic manner reflect these clarifications, while the majority of derivations are now in 'Quantification of cell uptake and monitoring cell mass and volume' section of Materials and Methods and in Supporting Information.
- e. Corrected the definition of $\epsilon = V_{\text{cell}}/V_{\text{sphere}}$, and not $\epsilon = V_{\text{sphere}}/V_{\text{cell}}$ as it was incorrectly written in the previous iteration of the manuscript.

2. Uncertainty estimations

The detailed uncertainty calculation for volume determination is questionable given that other key assumptions (e.g. the assumption of no impact of light refraction at the cell-medium interface on Line 40-41, and subsequent assumption of the RI increment of 0.18 mL/g on Line 155) were not quantified. Many aspects of the mathematical models of osmoregulation also include assumptions that are not quantified.

We have performed numerical simulations using the Beam Propagation Method described in the manuscript to quantify the effect of light refraction at the cell interface. We have now also explicitly stated in the manuscript that this uncertainty estimation is based on the properties of the optical system only, and that variations in other parameters may add to this uncertainty. Further, while we agree that there is an additional uncertainty in the results stemming from the chosen value of the refractive index increment $\left(\frac{dn}{dc}\right)$, this does not influence the uncertainty estimate of the cell volume, only of the cell mass. We have added a sentence about the uncertainty in $\left(\frac{dn}{dc}\right)$ in the introduction.

3. The pressure model

Line 219. The pressure relationship only holds at steady state.

Line 240-241. The estimate of osmotically susceptible volume should be compared to relevant literature. For example, a comprehensive discussion of the non-osmotic volume, literature references, and a model-based estimation from data are given in <https://www.ncbi.nlm.nih.gov/pubmed/16922683>. The estimate is 0.37, and hence the osmotically susceptible volume is 63% (which, notably, is very similar to the estimate (66%) in the present manuscript).

We have clarified that the osmotic pressure relation holds only at steady state, although the method itself can be applied, with suitable changes in the mathematical representation, also to non-equilibrium situations. We also thank the referee for pointing us to the work of Gennemark et al., which we now appropriately reference in connection with the estimate of the non-osmotic volume.

Figure 4A. It would be of interest to see time-resolved data for cell mass (volume is already shown in the inset). Data of this kind are of high interest to the modelling community to revise or refine current state-of-the-art models of yeast osmoregulation. The authors should add all relevant meta-information about the experiment (such as type, level and time-points of the added osmotic stress, and are recommended to add raw data to the supplementary material).

We have made considerable changes to the final figure of the manuscript, which we believe now both clarifies the measurement and analysis procedure, as well as providing additional information at the single cell level. Specifically we show the increase in cell mass as a function of time for a set of four cells, and estimate at the single cell level not only the molar mass of the produced osmolytes but also the production rate of these compounds, showing that cells under these experimental

circumstances produce approximately 0.5 mM osmolytes (likely mostly glycerol) per second, with a considerable cell-to-cell variability. Further, we found that small cells (smaller than about 30 fl) appear to produce heavier compounds, which may be attributed to simultaneous overall cell growth and osmoadaptation.

4. Perspective

How do the findings of a linear relationship between cell dry mass and cell volume (for medium to large cells) and of higher cytosol density in cells with volume smaller than 20 fL relate to previous data in the literature?

We have also added a reference to Loferer-Kröbächer et al., *Appl. Environ. Microbiol.* (1998), in which the ratio of dry mass to cell volume was found to be inversely related to bacterial size.

MINOR COMMENTS

Consistently insert a space between number and unit. There are several instances to correct, e.g. on lines 79, 87, 92, 125, 127, 129, 148, 149, 170, 298, 310, 312, and 328.

Line 40: effects...is -> effects...are

Line 45: is composed of -> depends on

Lines 86, 134 and 335: The latin name for Baker's yeast was given in the Introduction, no need to repeat.

Line 100: corresponds -> correspond

Line 111: what is meant by "numerical model cells"?

Line 116-117: Use the abbreviation BPM (just introduced).

Line 136: Most of this already described in Materials and methods.

Line 153: What is "obtained as": the correlation, the volume or the mass?

Line 174: space between "intra-" and "and".

Line 175: unclear what "which is" refers to.

Lines 175-179: This sentence is too long, very hard to interpret, and (likely) grammatically incorrect.

Line 215: Hohmann et al. -> Klipp et al.

Line 268: contnent -> content

Line 375: determine -> to determine

Figure 1A: Components in the figure are undefined.

Figure 2A: The time-points of solute exchange should be indicated.

Figure 2C: What is meant by the red dashed line in the inset?

Figure 2 legend: Inset to D not defined. Cannot see a red dashed line or a blue dotted line in Figure D.

Figure 3A: The time-points of solute exchange should be indicated. Which solutes were included?

Figure 3 legend. Define "brief".

Figure 4 legend. is shown -> are shown (Line 321)

We thank the referee for pointing us to these typos and grammatical inconsistencies. We have addressed them in the revised manuscript.

Reviewer #3:

The authors present a label-free imaging technique (digital holography microscopy) to visualize osmoregulatory response of individual yeast cells due to external perturbation of the local

environment.

The work is highly novel and would be of great interest to many researchers in the field of osmoregulation, mainly due to the low cost of the imaging system and low technical difficulties of rebuilding a similar system.

Even though the manuscript can be published as is, the image analysis part would benefit from a more detailed explanation why a shift to the center of the Fourier spectrum is necessary. To my knowledge this is not a maneuver that is commonly used, even though every pixel is treated equal in the process.

My recommendation is to accept the manuscript for publication.

The reviewer has identified a key element in the processing of the Fourier image. Physically, the two off-center peaks describe the optical field multiplied by a plane wave described by $\exp(\pm i\mathbf{k}_{\text{off}} \cdot \mathbf{x})$, where \mathbf{k}_{off} represents the off-axis angle of the optical setup. Thus, by shifting one of the off-center peaks to the center of the Fourier transform the effect of this plane wave can be eliminated and the optical field can be extracted. The same effect can be obtained by multiplying the hologram by the complex conjugate of the plane wave prior to taking the Fourier transform. A clarification statement to this effect has been added to the manuscript.

Reviewers' comments:

Reviewer #1 (Remarks to the Author):

This reviewer appreciate the efforts made by the authors for improving the manuscript.

However the main criticism on the suitability for Nature Communications is still confirmed for the revised manuscript as no enough aspects of novelty can be found in the work that justify the publication in a high rank journal.

Scientific Report is the most suitable choice for this work.

Reviewer #2 (Remarks to the Author):

The response letter made me eager to read the new submission.

However, when reading the revised manuscript, I was surprised to see that it had not been properly proof-read. I got stuck already in the Introduction (see comments below). Given that this a submission to a prestigious journal, and considering the comments from my side in the previous iteration, I had hoped for a much more mature manuscript.

Scanning the Materials and methods section confirms this observation (e.g. lines 146-147 and 177-178 with descriptions of how data were fitted to different models; I assume that a model was fitted to experimental data).

The authors are strongly recommended to carefully proof-read the manuscript before asking me to review it.

Comments to the Introduction

30: phase separated -> phase-separated

31: Unclear what "This" refers to

31: an protective -> a protective

33: It is unclear what "specific binding" means. Binding of what, or binding to what?

36: The term "cellular environmental responses" is ambiguous. Do you mean the cell's response to changes in the environment?

38: unclear what "which" refers to: population heterogeneity, or changes in morphology, or biophysical changes, or a combination of those.

40: For consistency, either add "cell" before "volume", or skip "cell" before "mass".

43: time resolved -> time-resolved

43: Would it be a challenge if it was resolved? A suggestion is to skip the word "unresolved".

53: The symbol Φ (Phi uppercase) has not been introduced.

53: Use a consistent notation for multiplication. In Eq 1, but not in Eq 2, a dot was used between factors.

54: Delta n, introduced on line 48, is denoted by difference in RI (unitless). On this line, 54, dn/dc (volume/mass) is introduced and confusingly referred to as RI increment. Clearly, Delta n and dn/dc differ as can be seen from the different dimensions. A more proper introduction of dn/dc is required. It may be worth pointing out that as RI is linearly related to c, dn/dc should be a constant.

56: "for biomolecules" is redundant

60: what is "this mass" referring to?

60: the variable m_w has not been introduced.

70-72: the sentence starting with "The latter" doesn't make sense. What is expressed, and what is

measured?

80-83: the sentence starting with "Further" is too long, and includes some ambiguity (what is "which" referring to?).

Response to Reviewers Comments

Label-free, spatio-temporal monitoring of cytosolic mass, osmolarity and volume, in living cells

Daniel Midtvedt, Erik Olsén, Fredrik Höök & Gavin D. M. Jeffries

We kindly thank the reviewers as well as the editor for their evaluation of this manuscript. Please find below the itemized responses to the comments and inserted manuscript statements. In conjunction with this document, please find a revised manuscript and supplementary information, containing all of the changes, and a separate document where all changes to the manuscript have been highlighted for ease of identification.

Referee #1:

This reviewer appreciate the efforts made by the authors for improving the manuscript.

However the main criticism on the suitability for Nature Communications is still confirmed for the revised manuscript as no enough aspects of novelty can be found in the work that justify the publication in a high rank journal.

Scientific Report is the most suitable choice for this work.

Upon receiving the referee reports, it has become clear to us that the novelty and implications of our results was not sufficiently emphasized in the manuscript. First, the ability to correlate stress responses of individual cells to relevant biophysical parameters, such as cell size, in a label-free manner provides researchers with a new tool to study how microorganisms cope with a changing extracellular environment. In particular, the quantitative single-cell approach is shown to enable new scientific questions to be addressed, such as how population heterogeneity affects proliferation of the population when faced with unfavorable conditions; a question which is very difficult to answer with traditional, ensemble-averaged methods.

Second, in this work we have demonstrated quantification of mass transport to and from micrometer sized biological compartments. A natural extension to this is quantification of mass transport to and from microscopic *intracellular* compartments, which would serve as a valuable complement to existing methods, both label-based and label-free. In the current revision of the manuscript we have better emphasized these unique aspects.

Thirdly, we have identified a means to characterize the compounds produced by cells in response to stress, in terms of their molar refractive index increment (the product of molar mass and specific refractive index increment). This demonstrates that one of the main obstacles in applying label-free techniques to biological systems, namely their lack of specificity, can under certain circumstances be overcome.

Referee #2:

The response letter made me eager to read the new submission.

However, when reading the revised manuscript, I was surprised to see that it had not been properly proof-read. I got stuck already in the Introduction (see comments below). Given that this a submission to a prestigious journal, and considering the comments from my side in the previous iteration, I had hoped for a much more mature manuscript.

Scanning the Materials and methods section confirms this observation (e.g. lines 146-147 and 177-178 with descriptions of how data were fitted to different models; I assume that a model was fitted to experimental data).

The authors are strongly recommended to carefully proof-read the manuscript before asking me to review it.

Comments to the Introduction

30: phase separated -> phase-separated

31: Unclear what "This" refers to

31: an protective -> a protective

33: It is unclear what "specific binding" means. Binding of what, or binding to what?

36: The term "cellular environmental responses" is ambiguous. Do you mean the cell's response to changes in the environment?

38: unclear what "which" refers to: population heterogeneity, or changes in morphology, or biophysical changes, or a combination of those.

40: For consistency, either add "cell" before "volume", or skip "cell" before "mass".

43: time resolved -> time-resolved

43: Would it be a challenge if it was resolved? A suggestion is to skip the work "unresolved".

53: The symbol Φ (Phi uppercase) has not been introduced.

53: Use a consistent notation for multiplication. In Eq 1, but not in Eq 2, a dot was used between factors.

54: Delta n, introduced on line 48, is denoted by difference in RI (unitless). On this line, 54, dn/dc (volume/mass) is introduced and confusingly referred to as RI increment. Clearly, Delta n and dn/dc differ as can be seen from the different dimensions. A more proper introduction of dn/dc is required. It may be worth pointing out that as RI is linearly related to c, dn/dc should be a constant.

56: "for biomolecules" is redundant

60: what is "this mass" referring to?

60: the variable m_w has not been introduced.

70-72: the sentence starting with "The latter" doesn't make sense. What is expressed, and what is measured?

80-83: the sentence starting with "Further" is too long, and includes some ambiguity (what is "which" referring to?).

In shaping up the resubmission our focus was primarily on reanalyzing the data surrounding the last figure of the manuscript, to highlight that the method can be used to correlate the rate of osmolyte accumulation and the *molar refractive index increment*, i.e. the product of molar mass (M) and specific refractive index increment (commonly denoted by $\left(\frac{dn}{dc}\right)$), to cell size at the level of individual cells. Unfortunately, and we apologize for this, this led us to overlook the linguistic aspect of the manuscript. We have for the current iteration made a thorough search for grammatical inconsistencies and, where necessary, reshaped some sections to clarify our reasoning.

REVIEWERS' COMMENTS:

Reviewer #2 (Remarks to the Author):

Minor comments

Eq. 4: Does this equation really follows straight-forward from the previous information given? If not, a motivation (e.g. "by combining Eq X and Y we get") or a literature reference to guide the reader would be appropriate.

Consistently insert a space between number and unit. There are several (>10) instances to correct.

109-111: check grammar

149: were->was

Figure 3F. It is not possible to distinguish circles from squares.

358: Motivation for the choice of trehalose as comparator is lacking

Response to Reviewers Comments

Label-free, spatio-temporal monitoring of cytosolic mass, osmolarity and volume, in living cells

Daniel Midtvedt, Erik Olsén, Fredrik Höök & Gavin D. M. Jeffries

We kindly thank the reviewers as well as the editor for their evaluation of this manuscript. Please find below the itemized responses to the comments and inserted manuscript statements.

Response to referee #2:

Eq. 4: Does this equation really follows straightforward from the previous information given? If not, a motivation (e.g. "by combining Eq X and Y we get") or a literature reference to guide the reader would be appropriate.

Eq. 3 follows from evaluating Eq. 2 at two different medium RI (n_1 and n_2). Eq. 4 then follows from Eq. 2 and Eq. 3. We have reformatted the equation to make this connection clearer.

Consistently insert a space between number and unit. There are several (>10) instances to correct.

We have reviewed the manuscript and inserted a space between number and unit where it was missing.

109-111: Check grammar

We have corrected this sentence.

149: were->was

We have corrected this mistake.

Figure 3F: It is not possible to distinguish circles from squares

We have replaced the circles, representing the cell volume and cell mass prior to osmotic upshift, by open diamonds which are clearly distinguishable from the squares which represent the cell volumes and cell masses after osmotic upshift.

358: Motivation for the choice of trehalose as comparator is lacking.

Trehalose is a compound that can be accumulated in the yeast cytoplasm at high concentrations under certain circumstances (such as heat stress and osmotic stress on cells grown on ethanol). We have clarified this choice of comparator in the manuscript.